# TIGERScore: Towards Building Explainable Metric for All Text Generation Tasks

♦**Dongfu Jiang**[*] , ♦**Yishan Li**[*], ♦**Ge Zhang**, ♡**Wenhao Huang**, ♦**Bill Yuchen Lin**, ♦**Wenhu Chen**
♦**University of Waterloo**, ♦**Tsinghua University**, ♡**01.AI**, ♦**Allen Institute for AI**
{dongfu.jiang,wenhuchen}@uwaterloo.ca, liyisha19@mails.tsinghua.edu.cn

Reviewed on OpenReview: https://openreview.net/forum?id=EE1CBKC0SZ

## Abstract

We present TIGERScore[1], a **T**rained metric that follows **I**nstruction **G**uidance to perform **E**xplainable, and **R**eference-free evaluation over a wide spectrum of text generation tasks. Different from other automatic evaluation methods that only provide arcane scores, TIGERScore is guided by natural language instruction to provide error analysis to pinpoint the mistakes in the generated text. Our metric is based on LLaMA-2, trained on our meticulously curated instruction-tuning dataset MetricInstruct which covers 6 text generation tasks and 23 text generation datasets. The dataset consists of 42K quadruple in the form of (instruction, input, system output → error analysis). We collected the 'system outputs' through from a large variety of models to cover different types of errors. To quantitatively assess our metric, we evaluate its correlation with human ratings on 5 held-in datasets, 2 held-out datasets and show that TIGERScore can achieve the open-source SoTA correlation with human ratings across these datasets and almost approaches GPT-4 evaluator. As a reference-free metric, its correlation can even surpass the best existing reference-based metrics. To further qualitatively assess the rationale generated by our metric, we conduct human evaluation on the generated explanations and found that the explanations are 70.8% accurate. Through these experimental results, we believe TIGERScore demonstrates the possibility of building universal explainable metrics to evaluate any text generation task.

## 1 Introduction

Evaluation for natural language generation tasks is a long-standing challenging problem. With the recent advancement of large pre-trained language models (Brown et al., 2020; OpenAI, 2023; Touvron et al., 2023), text generation models have become more widely adopted in real-world applications than ever. Newly developed text-generative models are being deployed across a wide range of downstream applications. As more and more people use these generative models, there is an increasing need to develop trustworthy evaluation metrics. Though GPT-4 has shown to achieve strong evaluation correlation with humans (Zheng et al., 2023), the open-sourced metrics are lagging significantly behind and mainly suffer from specific issues:

**Dependency on references**: Some evaluation metrics like ROUGE (Lin, 2004), BLEU (Papineni et al., 2002b), COMET (Rei et al., 2020), InstructScore (Xu et al., 2023c) would require gold references to measure the quality. These metrics compare the generated output against one or more reference texts to assign the evaluation score. However, such an assumption can be highly unrealistic in many downstream applications where the gold reference is hard to collect.

**Limited to specific domains**: Some evaluation metrics are limited to specific domains, lacking the ability to generalize to broader text generation tasks. For example, COMET (Rei et al., 2020), BLEURT (Sellam

---

[*]Dongfu Jiang and Yishan Li have led the project and contributed equally.
[1]Project website: https://tiger-ai-lab.github.io/TIGERScore/

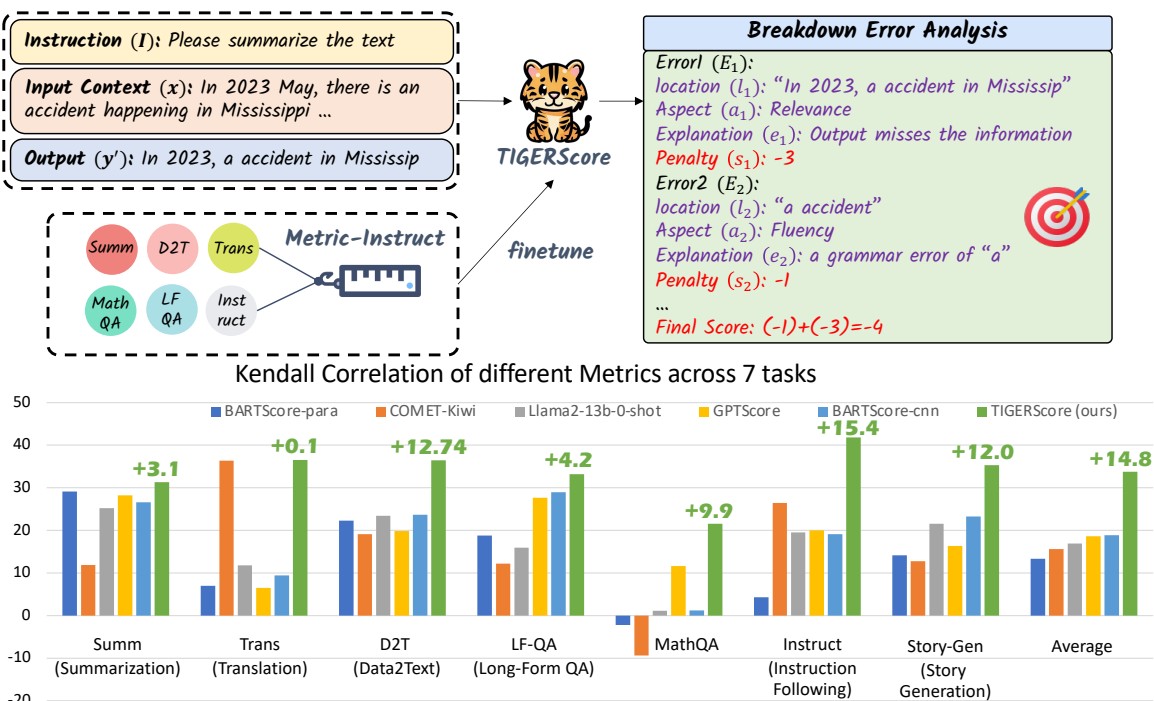

Figure 1: The upper part shows the input and output format of TIGERScore, which takes an instruction and an input context, along with the model-generated output that is to be evaluated as the inputs, and output detailed breakdown error analysis with explanations. It is finetuned on our curated dataset MetricInstruct. The lower part shows the Kendall's correlation of different metrics w.r.t human ratings, from which we can see TIGERScore achieves the SoTA correlation on most generation tasks.

et al., 2020), SESCORE2 (Xu et al., 2023b) and InstructScore (Xu et al., 2023b) are specifically designed for tasks like machine translation or WebNLG tasks.

**Lack of attribution:** Some evaluation metrics tend to directly output a score without any attributions, e.g., where the errors occur and why. For instance, BARTScore (Yuan et al., 2021), BERTScore (Zhang et al., 2019) and GPTScore (Fu et al., 2023) adopt the pre-trained language models' log likelihood as the evaluation metric. Such metrics do not provide the location and reason for the assigned score, which limits their trustworthiness and reliability.

To address these issues, we propose a novel metric, TIGERScore, a **T**rained metric that follows **I**nstruction **G**uidance to perform **E**xplainable and **R**eference-free evaluation. As shown in Figure 1, the input to TIGER-Score consists of an instruction describing the task definition, the task input, and the system output. TIGERScore is capable of generating breakdown error analysis that can (1) locate each mistake, (2) explain the mistake and suggest revisions, and (3) provide a penalty score (between [-5, -0.5]) for each mistake. The final score for the system output can be calculated by summing up all the penalty scores.

TIGERScore is built by fine-tuning LLMs on our curated MetricInstruct dataset, which contains a total of 42K examples of (instruction, input, system output, error analysis), obtained from 23 diverse text generation datasets. The dataset includes system outputs from more than 50 systems, covering a wide variety of errors. The error analysis is curated by prompting GPT-4 (OpenAI, 2023) and filtered through various strategies. The tuned model TIGERScore has shown the highest overall Kendall's correlation with human ratings on seven major text generation tasks as depicted in Figure 1. TIGERScore is highly convenient to use because it does not require any additional reference. We evaluate TIGERScore on 5 held-in datasets and 2 held-out datasets. As a reference-free metric, TIGERScore surpasses the best reference-free metrics (GPTScore (Fu et al., 2023)) by 15% and surpasses the best reference-based metrics (COMET-22 (Rei et al., 2022a)) by a margin of 8% in terms of Kendall's score. On the two held-out datasets, TIGERScore demonstrates unprecedented generalization capabilities. We further employ humans to evaluate the explanations

generated by TIGERSCORE and found that over 70% of the generated explanations are highly accurate and trustworthy. Our analysis shows that the success of TIGERSCORE is attributed to three key aspects in our curated `MetricInstruct`: (1) dataset diversity, (2) error coverage, and (3) high quality, which enable TIGERSCORE to generalize better on unseen tasks than any other metric.

## 2 TIGERScore

TIGERSCORE is built upon three design criteria: (1) It is driven by instructions, making it easily adaptable to any text generation task. (2) The evaluation process eliminates the need for a "gold standard" or perfect example for comparison. (3) It is highly explainable, as the model can generate an error analysis that helps the user understand each identified mistake and its associated penalty.

### 2.1 Background

The pursuit of improved metrics for text evaluation has been a significant focus since the inception of language models. Automatic n-gram-based metrics (Elliott & Keller, 2014; Callison-Burch et al., 2006; Isozaki et al., 2010) have always served as the default metric, computing the n-gram match F-1 score with the reference text until research highlighted their significant weaknesses in aligning with human preferences. Later, neural metrics were proposed to better capture the semantic similarity in addition to mere morphological similarity only by either computing based on neural representation (Zhang et al., 2019; Yuan et al., 2021) or directly fine-tuning with human preferences (Rei et al., 2020). It has also been demonstrated that multi-aspect scores, using the logits of large language models with well-designed instructions for various aspects as prompts (Fu et al., 2023), could achieve an even higher correlation.

There have been some attempts to build explainable metrics leveraging the great capacity of large language models. For instance, UniEval (Zhong et al., 2022a) constructs a multi-aspect evaluation system by individually training on aspect-specific data. PandaLM (Wang et al., 2023a) compares two responses for a given instruction and input to judge which is better, providing a short reason for its decision. InstructScore (Xu et al., 2023c) evaluates the quality of translation by training Llama-2 to compare the reference and translation, listing errors with structured information. However, none of these metrics has been able to address all the issues mentioned in section 1 concurrently.

### 2.2 Problem Formulation

Suppose $y'$ is the system output from a given source context $x$ with a specific natural language instruction $I$ to describe the task definition. And $y$ is a corresponding reference output. If a metric uses $y$, it's reference-based, otherwise, it's reference-free. For example, when $T$ refers to "translation", an instruction $I$ for that task could be "Translate the following text from Chinese to English". For each task type $T$, we ask the evaluation metric to focus on a few pre-defined evaluation aspects $A_T$ like relevance, factuality, fluency, etc.

TIGERSCORE is a reference-free metric, defined by a function $f$ to take the triple $(I, x, y')$ as input, to produce a list of structured errors $\{E_1, ..., E_m\}$ as the otuput, called the error analysis content:

$$\{..., E_i, ...\} = \{..., (l_i, a_i, e_i, s_i), ...\} = f(I, x, y') \tag{1}$$

where $(l_i, a_i, e_i, s_i)$ denotes specific information of the error $E_i$. Specifically, $l_i$ points to the location of the error, and $a_i \in A_T$ is a pre-defined aspect to which this error belongs. $e_i$ comprises both the explanation of this error and its revision suggestion. $s_i$ is the penalty score reduced for this error, which lies in $[-5, -0.5]$. The final score of $y'$ is computed as the sum of all the penalty scores: $s_{y'} = \sum_i s_i$. The range of the final score lies within $(-\infty, 0]$, where 0 means perfect generation and a lower score indicates worse quality and more severe errors. However, in practice, the number of detected errors is affected by both the training data and the generation length, thus leading to less than 10 errors most of the time during the evaluation.

### 2.3 Multi-Aspect Evaluation

As stated in the problem formulation, each error is considered to be attributed to a certain aspect that the system output might make mistakes on. For each task, 3 or 4 aspects are designed with the help of both

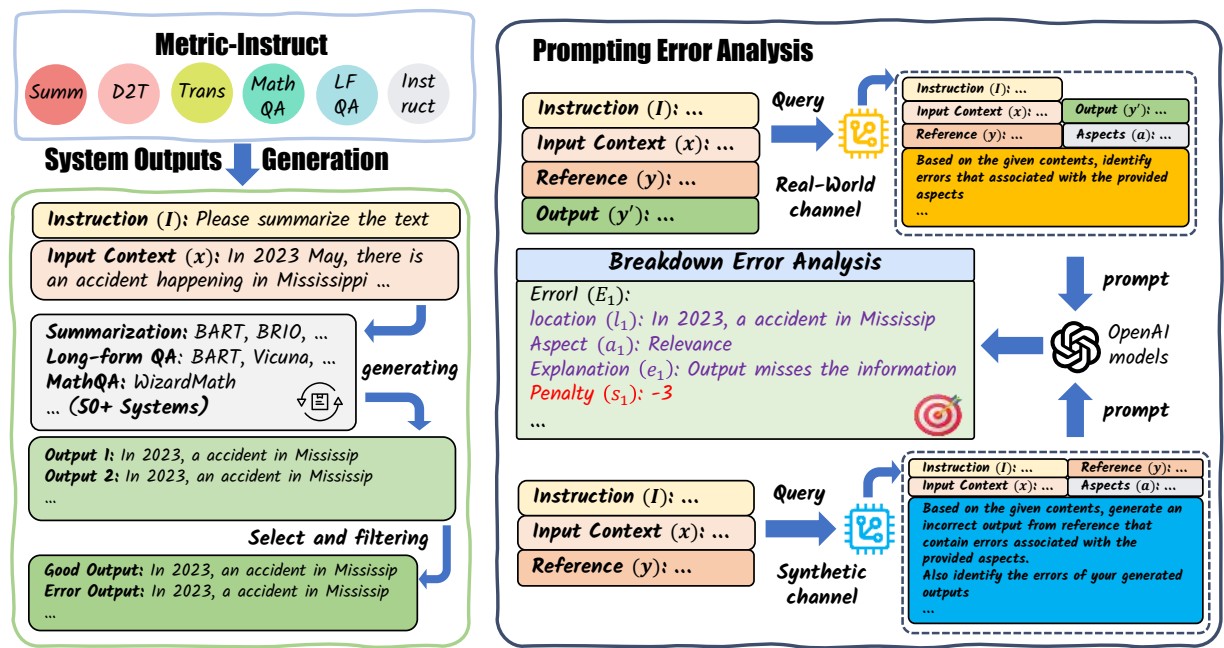

Figure 2: Overall pipeline of constructing MetricInstruct through the two-channel collection. The main difference is that the real-world channel collects outputs $y'$ from real-world models, while the synthetic channel asks GPT-4 to synthesize an error output based on the provided reference.

GPT-4 prompting and human supervision, detailed in subsection A.6. These aspects are designed to be both non-overlapping (mutually exclusive) and collectively exhaustive. For instance, 4 aspects to evaluate an output for the instruction-following tasks is shown in Table 1.

## 2.4 Training Setup

TIGERSCORE is finetuned on Llama-2-7B and Llama-2-13B (Touvron et al., 2023) respectively with both's batch size being 128. The prompt templates used for both the fine-tuning and inference are given in Table 9. The model's maximum context length is set to 1024. We use a cosine learning scheduler with the warmpup ratio being 0.1 We finetune the 7B version on 4 A100 GPUs (80GB) for 3 epochs with a learning rate of 2e-5. And 13B version is run on 8 A100 GPUs (80GB) for 2 epochs with a learning rate of 1e-5. Inference of TIGERSCORE is conducted on a single A100 GPU with the assistance of the VLLM toolkit to increase the speed. (Kwon et al., 2023)

| Instruct | |
|---|---|
| Aspect | Definition |
| Comprehension | Evaluates how well the output understands the given instruction. |
| Accuracy | Measures the correctness of the output in relation to the instruction and the paired input context. |
| Informativeness | Assesses the relevancy and usefulness of the information provided by the output. |
| Coherence | Evaluates how logically the output flows and connects. |

Table 1: Definitions of evaluation aspects of TIGERSCORE for Instruction-following (Instruct) as an example. See full table in Table 10 for aspect definitions of all 6 text generation tasks

| Task | Real-World (training set) | | | Synthetic (training set) | | |
|---|---|---|---|---|---|---|
| | Dataset | Output Source | # Sample | Dataset | Output Source | # Sample |
| Summarization (Summ) | SummEval‡, XSum, Newsroom,SAMSum | 27 Systems | 4339 | CNN/DM,XSum, Gigaword,SAMSum | GPT-4 | 612 |
| Translation (Trans) | WMT-22‡ | 18 Systems | 5507 | WMT-22 | GPT-4 | 672 |
| Data2Text (D2T) | WebNLG-2020‡ WikiTableText,ToTTo | 17 Systems | 4701 | WikiTableText Dart,ToTTo | GPT-4 | 160 |
| Long-Form QA (LF-QA) | ASQA,FeTaQA CosmosQA,ELI5 | 5 Systems | 3370 | ASQA,FeTaQA Cosmos QA,ELI5 | GPT-4 | 2146 |
| MathQA | GSM8K | 5 Systems | 4529 | None | | |
| Instruction-Following (Instruct) | MixInstruct‡ | 11 Systems | 2248 | AlpacaFarm,Dolly Guanaco,OASST | GPT-4 | 3014 |

Table 2: The composition of our dataset. For synthetic data, the output is generated by asking GPT-4 to synthesize incorrect outputs that contain a few specific types of errors. For the datasets with ‡, we take their pre-released system outputs. For the others, we collect the system outputs on our own.

## 3  MetricInstruct

We present the `MetricInstruct` dataset, which is employed to fine-tune TIGERSCORE. The three underlying criteria for dataset construction are: (1) **dataset diversity**: we choose 23 distinctive datasets as the source context to cover enough generation tasks. (2) **error coverage**: we take system outputs generated from 50+ text generation systems to cover all types of errors and guarantee a balanced distribution. (3) **quality ensurance**: to ensure `MetricInstruct` is tailored to gather in-depth error analysis as detailed in subsection 2.2, we sourced it by prompting GPT-4 (OpenAI, 2023) and then filtered through different heuristics to eliminate low-quality error analysis.

### 3.1  Diverse Dataset Source

`MetricInstruct` incorporates samples from 23 distinctive text generation datasets, which are categorized into 6 major categories of text generation tasks. While the collection encompasses well-researched tasks such as summarization (Summ), translation (Trans), and data2text (D2T), it also introduces popular new tasks like Long-Form QA (LF-QA), MathQA, and instruction-following (Instruct). These latter tasks have witnessed limited evaluation research. Although the assessment of traditional tasks has dominated the research landscape, we posit that effectively evaluating these new tasks is crucial for constructing a comprehensive evaluator for all text generation domains.

In addition, we meticulously selected datasets for each task to ensure diverse coverage across the knowledge domain, as shown in Table 2. For instance, in the case of the LF-QA task, we utilize ASQA (Stelmakh et al., 2022b) to include knowledge about ambiguous factoid questions, while FeTaQA (Nan et al., 2022b) is employed to handle challenges related to tabular source question answering. We then selected samples from each dataset's training set, following the predefined constraints like maximum input and reference output lengths, topics balancing of instances, and so on.

### 3.2  Broad Coverage of Errors

As shown in Figure 2, our "system outputs" come from two channels, namely real-world system outputs and synthetic incorrect outputs, representing 2 main components of `MetricInstruct`.

We consider a wide range of systems and use their outputs as our evaluation input, as shown in Table 2. These outputs are either collected from pre-released outputs, like WMT-2021 official evaluation systems, or generated by us by prompting existing domain-specialized models, like BRIO (Liu et al., 2022), Wizard-Math (Luo et al., 2023), etc.

As illustrated in Figure 2, for those self-generated outputs, we sample 5 different outputs using top-p sampling for each instance using various output systems. After that, we use BARTScore to sort the 5

outputs. Outputs with lower BARTScore are heuristically considered to contain more errors and will be preferred to be chosen as the final hypothesis output $y'$ to be evaluated.

Subsequently, we utilized meticulously designed prompting templates (see in A.7) to elicit standardized error analysis from GPT-4. The main idea is to provide them with the instruction $I$, input $x$, the reference output $y$, and system output $y'$ along with the definitions of pre-defined aspects $A_T$, to query the OpenAI models to generate the list of errors, as described in subsection 2.2. We also report their correlation performance in Table 4, showcasing the reference-based ChatGPT results.

**Synthetic:** While real-world system outputs ensure the error distribution is aligned with practical scenarios, they might be overly tailored to the bias of these limited systems and omit certain error cases their outputs fail to cover. Therefore, we decided to synthesize more incorrect outputs that can balance under-represented errors by prompting GPT-4 with our designed templates.

Firstly, to complement the real-world system outputs to cover broader error cases, we prompt GPT-4 to deliberately generate designated erroneous outputs, modified from the existing reference output $y$, using the prompt template in Table 18. By supplying GPT-4 with a combination of randomly selected aspects and their definitions $A'_T$, we control the aspect of errors it produces so that error aspects can be more balanced. Datasets used in this step are reported in the right column of Table 2.

Secondly, to improve TIGERSCORE's generalization capability in evaluation for tasks or errors it has not previously encountered, we employ a strategy that involves prompting GPT-4 to generate data with customized error aspects tailored to individual instructions. Our rationale is that while the manually designed aspects are logical, they may not be specific enough for tasks that involve following instructions where various types of instructions will be included. Besides, this can also prevent the trained model from being overfitted to the pre-defined aspects in subsection A.6. To do this, we supplement the first part of synthetic data by sampling 10k data points from the Alpaca-52k dataset and using the template in Table 19 to prompt GPT-4 to get this second part of synthetic data.

### 3.3 Heurstic-based Filtering

We refine our preliminary raw dataset drawn from both **real-world** and **synthetic** channels using heuristic filtering techniques. Initially, we remove any anomalous data in JSON format. Instances marked by hallucinated or mismatched error locations, illogical severity labels, or excessively long outputs are considered flawed and excluded. Furthermore, given the reference-free nature of TIGERSCORE, an explanation that relies on reference outputs for justification is excluded. These steps eliminate approximately 35% of the initial data. Subsequently, we employ GPT-4 to assess the reasonableness of our error analysis, utilizing the template outlined in Table 20. This process further filtered out about 15% percentage data. This phase further excludes about 15% of the data. Ultimately, we compile a high-quality dataset of 42,484 instances, designated as `MetricInstruct`.

## 4 Experiments

### 4.1 Evaluation Datasets

We have gathered both the held-in and held-out datasets to compare the performance of TIGERSCORE with the existing baseline metrics. The basic statistics of some main datasets for each task are shown in Table 3. System outputs to be evaluated of each test dataset are either from official releases, such as SummEval (Fabbri et al., 2021), WebNLG-2020 (Zhou & Lampouras, 2020), WMT-22 (Freitag et al., 2022), and OpenMEVA (Guan et al., 2021), or by generating by ourselves, such as A-F-E-C (Stelmakh et al., 2022a; Nan et al., 2022a; Fan et al., 2019; Huang et al., 2019), GSM8K (Cobbe et al., 2021), LIMA (Zhou et al., 2023) and AlpacaEval (Li et al., 2023).

Human preference scores are necessary to conduct the correlation analysis. To get the gold preference scores we used to conduct the evaluation experiments, we collected either existing human ratings like WMT-22

| Task | Eval Dataset | Output Source | # Inputs | # Samples |
|------|-------------|---------------|----------|-----------|
| Held-in Evaluation Dataset (test set) | | | | |
| Summ | SummEval | 16 Systems | 100 | 1600 |
| Trans | WMT-22 (zh-en) | 18 Systems | 1875 | 33750 |
| D2T | WebNLG-2020 | 18 Systems | 179 | 2848 |
| LF-QA | ASQA, FeTaQA, CosmosQA, ELI5 | 4 Systems | 400 | 1600 |
| Math QA | GSM8K | 2 Systems | 1319 | 2638 |
| Held-out Evaluation Dataset (test set) | | | | |
| Instruct | LIMA,AlpacaEval | 9 Systems | 500 | 4500 |
| Story-Gen | OpenMEVA (ROC) | 5 Systems | 200 | 1000 |

Table 3: Overview of our main evaluation datasets. "# Samples" is computed by multipling "# Inputs" with the number of outputs systems, representing the total instances that a metric need to compute.

MQM scores for translation, or GPT-4 rated scores for Long-form QA. Details of the curation are shown in subsection A.5.

## 4.2 Baselines

We categorize the baselines into reference-based and reference-free metrics for fair comparison.

**Reference-based:** We choose popular metrics, including BLEU (Papineni et al., 2002a), ROUGE (Lin, 2004), BERTScore (Zhang et al., 2019), BLEURT (Sellam et al., 2020) and BARTScore (Yuan et al., 2021). Recent emerging metrics are also included, like COMET-22 (Rei et al., 2022a),UniEval (Zhong et al., 2022b), GPTScore (Fu et al., 2023), and InstructScore (Xu et al., 2023c). Specifically, we use BARTScore-ref to denote that we adopt the ref-hypo scoring type. For GPTScore, we use FLAN-T5-base (Chung et al., 2022) as base models and use GPTScore-ref to denote that we adopt the f-1 average of the ref-hypo and hypo-ref scores. We also report the zero-shot results of GPT-3.5-turbo with the same prompting templates as those used for generating real-world channel data. Reporting this baseline will help us understand whether TIGERScore has surpassed its easy substitute using the cheap OpenAI model, thus proving its effectiveness.

**Reference-free:** We choose BARTScore, GPTScore and COMETKiwi (Rei et al., 2022b) as reference-free baselines to compare. Specifically, we use BARTScore-src to denote the src-hypo scoring type, thus making it a reference-free metric. For GPTScore, we still use the FLAN-T5-base model and use GPTScore-src to denote that we use the src-hypo scoring type. We also include the frequently used 0-shot results of Llama-2-13b (Touvron et al., 2023) and GPT-4 (OpenAI, 2023)for better comparison and to know more about the performance gaps with GPT-4.

## 4.3 Main Results

We present a comprehensive analysis of TIGERScore across all 5 held-in tasks and 2 held-out task in Table 4 reporting Kendall correlation results. Additionally, we provide supplementary results on Pearson and Spearman correlations in the Appendix (see Table 25 and Table 26). We average the performance across these tasks to gauge the general ability of the model.

Our results highlight the significant advantages of TIGERScore over other reference-free metrics. Notably, it has surpassed all other reference-free metrics in Kendall correlation. In Pearson correlation, it is the highest for 6 out of 7 tasks. This underscores the robustness and consistency of TIGERScore in evaluating text generation tasks.

Compared with reference-based baselines, TIGERScore generally outperforms most reference-based metrics. However, it does score lower than some task-specific metrics like UniEval (summ) in summarization,

| Tasks→ | Summ | Trans | D2T | LF-QA | MathQA | Instruct | Story-Gen | Average |
|---|---|---|---|---|---|---|---|---|
| GPT-based Metrics | | | | | | | | |
| GPT-3.5-turbo (zero-shot) | **30.45** | 32.30 | 30.38 | 20.91 | **58.57** | 17.73 | 3.26 | 27.65 |
| GPT-4 (zero-shot) | 29.32 | **35.38** | **32.26** | **35.85** | 46.63 | **49.50** | **25.69** | **36.38** |
| Reference-based Metrics | | | | | | | | |
| BLEU | 8.71 | 14.50 | 23.13 | 7.73 | 17.25 | 35.92 | -0.89 | 15.19 |
| ROUGE-2f | 10.67 | 13.19 | 24.74 | 11.73 | 18.07 | 34.59 | 1.78 | 16.40 |
| InstructScore | 20.86 | 40.44 | 30.21 | 15.64 | -3.87 | 13.87 | 13.50 | 18.66 |
| GPTScore-ref | 10.80 | 18.74 | 27.47 | 22.13 | 14.86 | 25.40 | 12.78 | 18.88 |
| BARTScore-cnn (hypo-ref) | 10.00 | 21.06 | 27.04 | 20.67 | **19.07** | 24.70 | 18.58 | 20.16 |
| BARTScore-para (hypo-ref) | 10.41 | 24.90 | 28.42 | 20.24 | 14.10 | 26.13 | 12.11 | 19.47 |
| BERTScore | 17.39 | 31.57 | 30.74 | 17.70 | 9.41 | 35.61 | 2.00 | 20.63 |
| BLEURT | 12.69 | 36.12 | **34.48** | 23.11 | 2.88 | 27.94 | 19.18 | 22.34 |
| UniEval (summ) | **35.89** | 16.08 | 28.56 | **29.32** | 16.15 | 11.93 | **31.22** | 24.17 |
| COMET-22 | 25.01 | **42.79** | 23.43 | 24.66 | -4.52 | **36.17** | 27.52 | **25.01** |
| Reference-free Metrics | | | | | | | | |
| BARTScore-para (src-hypo) | 29.12 | 7.01 | 22.32 | 18.80 | -2.21 | 4.26 | 14.15 | 13.35 |
| BARTScore-cnn (src-hypo) | 26.63 | 9.40 | 23.69 | 28.93 | 1.23 | 19.09 | 23.29 | 18.89 |
| Llama-2-13b-chat-0-shot | 25.22 | 11.79 | 23.45 | 15.96 | 1.08 | 19.50 | 21.52 | 16.93 |
| COMETKiwi | 11.87 | 36.37 | 19.08 | 12.23 | -9.38 | 26.46 | 12.78 | 15.63 |
| GPTScore-src | 28.20 | 6.50 | 19.81 | 27.64 | 11.64 | 20.04 | 16.36 | 18.60 |
| TIGERScore-7B | 28.79 | 33.65 | 32.44 | 33.93 | 19.98 | 38.13 | 29.72 | 30.95 |
| TIGERScore-13B | **31.29** | **36.50** | **36.43** | 33.17 | **21.58** | **41.84** | **35.33** | **33.73** |
| Δ (ours - best reference-free) | +2 | +0 | +13 | +4 | +10 | +15 | +14 | +15 |
| Δ (ours - best reference-based) | -4 | -6 | +2 | +4 | +2 | +5 | +4 | +8 |

Table 4: The Kendall correlation results of all the baseline metrics and TIGERScore on the evaluation datasets shown in Table 3. For each task, the metric with the highest correlation to average performance is highlighted in bold. The results of Pearson and Spearman are reported in Table 25 and Table 26 respectively, showcasing the same great performance of TIGERScore

COMET-22 in translation. We consider these discrepancies acceptable, because the compared metrics are reference-based and all specifically fine-tuned for a single task. Furthermore, TIGERScore achieves significantly higher overall correlation, compared with the cheap API substitute GPT-3.5-Turbo (zero-shot) and the Llama-2-13b-chat (zero-shot), proving its effectiveness. What's exciting to note is that TIGERScore-13b can achieve comparable correlation performance with GPT-4 (zero-shot), even higher on summarization, translation, data2text, and story generation.

### 4.4 Human Evaluation

To better understand the quality of the generated error analysis provided by TIGERScore, a random selection of 50 error analyses from each evaluation dataset was assessed by human experts who rated them from the following perspectives: 1) Reasonableness, 2) Comprehensiveness, 3) Effectiveness, 4) Overall Estimation, whose definitions are given in the following:

**Reasonableness:** The human experts directly pointed out which errors are problematic in error analyses, examining whether the analysis contained hallucination or illogical reasoning.

**Comprehensiveness:** The human experts carefully review the source, output, and error analyses to determine if there are any additional errors unnoticed by TIGERScore. Based on human experts' analysis, they give a score on a scale of 1 to 4, specifically focused on identifying potential errors that may have been overlooked in the original analysis conducted by TIGERScore.

| Aspects | Explanation Error? | | Overlooked Errors | | | | Revision Suggestions | | | | | Overall Rating | | | | |
|---|---|---|---|---|---|---|---|---|---|---|---|---|---|---|---|---|
| Rate→ | No | Yes | 1 | 2 | 3 | 4 | 1 | 2 | 3 | 4 | 5 | 1 | 2 | 3 | 4 | 5 |
| Summ | **70** | 35 | 2 | **17** | 15 | 16 | 6 | 4 | **19** | 7 | 14 | 3 | 10 | **17** | 7 | 13 |
| Trans | **54** | 25 | 3 | 8 | 17 | **22** | 2 | 7 | 17 | 6 | **18** | 3 | 6 | 15 | 9 | **17** |
| D2T | 19 | **21** | 1 | 8 | 10 | **31** | 11 | 8 | 9 | 3 | **19** | 11 | 10 | 4 | 7 | **18** |
| LF-QA | **42** | 19 | 4 | 10 | 11 | **25** | 5 | 8 | 14 | 7 | **16** | 6 | 8 | 10 | 6 | **20** |
| MathQA | **39** | 26 | 5 | 12 | 12 | **21** | 5 | 7 | **19** | 5 | 14 | 4 | 9 | 10 | 13 | **14** |
| Instruct | 5 | **9** | 5 | 5 | 8 | **32** | **21** | 3 | 5 | 2 | 19 | 9 | 4 | 3 | 7 | **27** |
| Story-Gen | **66** | 29 | 7 | **16** | 13 | 14 | 7 | 6 | **16** | 10 | 11 | 7 | **12** | 11 | 9 | 11 |
| Total | **295** | 164 | 27 | 76 | 86 | **161** | 57 | 43 | 99 | 40 | **111** | 43 | 59 | 70 | 58 | **120** |

Table 5: Human evaluation results, the first question is asked per error in error analyses, and the others are per sample. Superior performance is indicated by higher numerical values. The most-voted rate of each task for each human evaluation aspect is bolded.

**Effectiveness:** The revision suggestions in error analyses are evaluated by human experts, on a scale of 1 to 5, to determine their appropriateness and effectiveness in enhancing the output quantity.

**Overall:** The Human experts further assign an overall score on a scale of 1 to 5 based on the reasonableness, comprehensiveness, and effectiveness of the error analysis.

We report the detailed human evaluation results in Table 5, it is found that 64.3% of TIGERSCORE's error analyses are deemed reasonable, that is, the answer to the first question is "no errors in interpretation". This suggests that most error analyses accurately identified and explained errors. In 70.6% of cases, evaluators gave a positive score (3 or 4) for question 2, implying no missing errors were found. This demonstrates TIGERSCORE's effectiveness in comprehensive error analysis. Overall, 70.8% of error analyses received positive ratings (3 to 5), indicating good quality and usefulness in identifying and explaining errors according to human experts.

### 4.5 Hallucination Analysis

**Alignments on no-error outputs** In order to understand the hallucinations generated by TIGERScore, we used TIGERScore to evaluate the gold reference, which is used in reference-based metrics, and expect TIGERScore not to hallucinate errors on these no-error instances. Results are shown in Table 6.

In tasks related to instruction-following and long-form QA, TIGERScore demonstrates a high level of accuracy, avoiding hallucinations in over 85% of cases. This highlights its proficiency in producing factual and error-free analysis. However, TIGERScore is less consistent in tasks like summarization, translation, and data-to-text conversion. In these areas, it often fails to achieve perfect scores (0), but still frequently identifies gold references as either flawless or only minimally flawed (with score reductions less than 2). This could be due to the subjective nature of these tasks, where minor errors, such as the substitution of similar words, maybe more open to interpretation. Additionally, TIGERScore faces challenges in tasks like MathQA and story generation. These difficulties may stem from the inherent complexity of MathQA problems and the subjective nature of story creation, as well as specific limitations of TIGERScore in these areas. Improving TIGERScore's performance in these challenging tasks remains a topic for future research.

**Hallucinations analysis of TIGERScore outputs** To better understand how TIGERScore handles hallucinations, we conducted experiments across six different tasks. For each task, we ran TIGERScore-13B on 20 samples with errors in the system output. We then used GPT-4 to determine if these samples contained hallucinations or factual inaccuracies, as outlined in the prompting templates found in Table 22. According to the results in Table 7, approximately 89.28% of TIGERSCORE's error analyses are free from hallucinations or factual errors. We acknowledge the limitations of our study, including the small sample size and the reliance on GPT-4 rather than human evaluators. Nonetheless, our findings are significant, demonstrating that TIGERSCORE is effective at avoiding hallucinations in generated content.

| Tasks→ | Summ | Trans | D2T | LF-QA | MathQA | Instruct | Story-Gen | Average |
|---|---|---|---|---|---|---|---|---|
| Gold reference's score = 0 | | | | | | | | |
| TIGERScore-7B | 16.00 | 3.57 | 45.51 | 84.75 | 34.36 | 73.98 | 34.00 | 41.74 |
| TIGERScore-13B | 48.00 | 21.01 | 23.03 | 94.50 | 25.28 | 86.38 | 46.00 | 49.17 |
| Gold reference's score >-2 | | | | | | | | |
| TIGERScore-7B | 68.00 | 72.48 | 78.65 | 84.75 | 34.36 | 92.48 | 34.00 | 66.39 |
| TIGERScore-13B | 97.00 | 83.63 | 94.94 | 94.50 | 25.28 | 96.14 | 49.00 | 77.21 |

Table 6: TIGERScore's score on the gold reference of the test set. For each task, the 0 column refers to the percentage that TIGERScore reduces 0 scores for the gold references. The > −2 column refers to the percentage that TIGERScore reduces less or equal to 1 score on the gold references.

| Tasks | Summ | Trans | D2T | LF-QA | MathQA | Instruct | Story-Gen | Total |
|---|---|---|---|---|---|---|---|---|
| Accuracy | 95.00 | 95.00 | 90.00 | 85.00 | 90.00 | 75.00 | 95.00 | 89.28 |

Table 7: The accuracy of the error analysis from TIGERScore-13B assessed by GPT-4 that do not contain hallucinations or factual errors. if includes hallucinations or factual errors, assessed by GPT4.

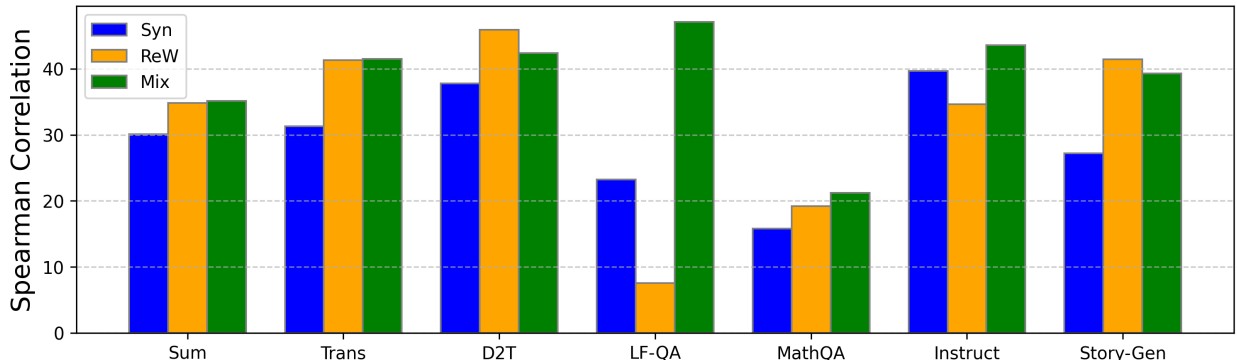

Figure 3: Investigation of the influence of Real-World & Synthesic mix training on the 7B model.

### 4.6 Abaltion Study on Data Source

**Ablation of Two Channel Data Collection**   To assess the effectiveness of our two-channel data in enhancing TIGERSCORE's performance, we fine-tune 3 models using different dataset setups for experiments: using only Real-World data (**ReW**), using only Synthetic data (**Syn**) and using the mixed of both (**Mix**). The results are detailed in the Spearman correlation in  Figure 3. the **Mix** model outperformed both **ReW** and **Syn** in 5 of the 7 tasks. It also achieves an average correlation approximately 20% higher than **ReW** and 31% higher than **Syn**. Slight decreases in correlation for D2T and Story-Gen tasks are deemed as acceptable compromises for better overall performance.

**Ablation of Each Single Generation Task**   In order to investigate the contribution of each task in the MetricInstruct, we conducted experiments to see whether multi-task training will benefit the single task. We trained a model for each task and compared its performance to that of TIGERSCORE. As depicted in the Table 8, the multi-task learning paradigm does benefit almost all tasks' performance, except for MathQA. We contend that Math QA poses a significant challenge for LLMs, and a separately trained model is more adept at handling this task.

| Metrics↓ Tasks→ | Summ | Trans | D2T | LF-QA | MathQA | Instruct | Average |
|---|---|---|---|---|---|---|---|
| Single Task | 35.55 | 41.65 | 39.75 | 41.60 | **26.75** | 41.19 | 37.75 |
| Multi Task | **36.81** | **44.99** | **45.88** | **46.22** | 23.32 | **47.03** | **40.71** |
| Δ (%) | 3.56 | 8.02 | 15.43 | 11.09 | -12.83 | 14.18 | 7.84 |

Table 8: Ablation of the influence of multiple tasks mix or single task on the 13B model.

# 5 Related Work

## 5.1 Instruction-driven Large language models

Instruction tuning has recently become the standard to "align" language models with more useful objectives and human preferences. The instruction tuning step is normally done to enhance certain skillset of large language models. Previously, instruction tuning has been focused on activating models' general capabilities to follow instructions to solve general tasks. Some work has been published like NaturalInstruction (Wang et al., 2022), FLAN (Wei et al., 2021) and T0 (Sanh et al., 2021) are the earliest work in the field. Later on, FLAN-v2 (Chung et al., 2022) have been proposed to understand the effect of scaling up the instruction datasets to understand its impact on model performance. These approaches mainly adopt human-annotated datasets to build the instruction following dataset. More recently, multiple works (Wang et al., 2023b; Xu et al., 2023a) propose to utilize synthetic instruction following data distilled from GPT-4 to align open-source LLMs. Our work differs from them in the sense that our method aims to activate specialized capability to generate error analysis according to instruction, which is the first of its kind.

## 5.2 Explainable Metrics

The increasing focus on model interpretability has led to a surge in research dedicated to explainable metrics. Research in these fields aims to build a metric system for a certain task that is readable to humans and is expected to help the development of better text generation systems (Leiter et al., 2022). Early endeavors in this area delved into explainability via multi-faceted evaluations, as exemplified by works such as Unieval (Zhong et al., 2022b) and GPTScore (Fu et al., 2023). As LLM blooms, researchers began to directly prompt LLMs to create interpretable metrics. One instance is PandaLM, trained on Llama to compare two responses pairwisely and provide a textual rationale for its decisions (Wang et al., 2023a). Another noteworthy approach is InstructScore, leveraging large language models as knowledge repositories to obtain premium error analysis examples (Xu et al., 2023c). Despite these commendable advancements, most existing explainable metrics still require gold references and are often limited concerning the task domain. Our contribution distinguishes itself by offering a reference-free nature and the cross-task ability brought by instruction-tuning over large language models, aiming to serve as a universal explainable metric.

# 6 Conclusion

In this paper, we propose the novel metric TIGERSCORE, which is able to evaluate any text generation task guided by natural language instruction. We demonstrate the exceptional performance of TIGERSCORE by its high correlation with human preference. We also demonstrate the high accuracy of its generated rationale. However, TIGERSCORE does hallucinate sometimes to generate false explanations. On the other hand, we found that TIGERSCORE is not good in evaluating reasoning tasks like mathQA. In the future, we plan to devote more effort to enable more faithful explanations and unleash its potential to evaluate errors for harder tasks, like reasoning errors.

# Limitation

**Hallucinated Errors** Despite substantial efforts to minimize hallucinations in TIGERScore's output, we still observe hallucinated errors, particularly in challenging tasks such as mathQA. This issue is attributed to both the quality of our training data and the limitations of our base model. A potential solution involves

initial fine-tuning on specific tasks for generation purposes, followed by further fine-tuning for evaluation. However, additional strategies are necessary to effectively reduce these hallucinations.

**Evaluation Efficiency**  TIGERScore, fine-tuned on the 7B and 13B versions of Llama-2, faces challenges with inference speed when used as an evaluation metric. Our testing reveals that TIGERScore achieves an evaluation speed of approximately 0.2 seconds per instance on a single A800 GPU with the assistance of VLLM. While this is manageable in interactive environments, further improvements are needed for efficiency in large-scale batch evaluations, compared to faster traditional metrics like BLEU and BERTScore.

**Discrepancy Between the Local Errors and Global Evaluation**  Dividing output evaluation into multiple local errors is logical, but using a simple summation of these errors as a global evaluation metric can lead to discrepancies. Longer outputs often have multiple errors, while shorter ones might be simply judged as entirely erroneous in a single error. Compared to global evaluation methods, like rating a score out of 10, developing a structured and reasonable method to accumulate and represent these errors remains an area for further exploration.

## Ethics Statements

Our work collected data from publicly available datasets that were ethically curated with informed consent from all participants. We ensure that all privacy data is excluded. We acknowledge the potential of our machine-learning models to generate hallucinated, biased, or unfair content. Methods have been adopted to prevent the generation of these kinds of content with our best efforts. Our research involves human evaluation experiments and we ensure that each participant's privacy is excluded and protected on our side. We make sure each participant is paid fairly according to their work amount. Our hourly rate is 12 dollars, which is above the US lowest payment rate.

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

# A  Appendix

## A.1  Prompting Strategies

In our study to extract high-quality error-analysis insights from GPT-4, we employed various intuitive prompting strategies. These strategies are detailed in the prompting templates found in subsection A.7. Key strategies are outlined below:

**Two-Step Generation and Formatting Process**   One of the challenges in eliciting structured knowledge from LLMs is directing them to generate content in a specific format. Although GPT-4 often adheres to given formats, there are instances of deviations. We observed that enforcing a strict format can compromise content quality, as indicated by reduced correlation in our analysis. Our approach involves initially allowing GPT-4 to generate responses freely in the first conversation turn. In the subsequent turn, we request GPT-4 to reformat its response according to a predefined format. The initial generation templates for each task, along with a singular template for the formatting step, are listed in subsection A.7.

**Incorporation of Task-Specific Words in Initial Queries**   To leverage GPT-4's task-specific knowledge, we designed varied prompting templates for different tasks with slight modifications. Keywords like 'Source,' 'Instruction,' 'Reference,' 'Output,' 'Solution,' 'Translation,' and 'Summary' are dynamically utilized in various task contexts. In tasks like instruction-following, where the context is self-explanatory, we omitted specific keywords.

**Integration of Predefined Aspect Definitions for Focused Evaluation**   Directly requesting GPT-4 to evaluate task outputs often led to low-quality error identification. It either points out simple discrepancies with the reference or misses crucial evaluation aspects, thus overlooking some errors. To address this, we incorporated predefined evaluation aspects defined in Table 10 into the templates, guiding GPT-4 to produce more focused responses. Exceptionally, for data2text, we found that directly evaluating errors was good enough, and thus, we did not include our predefined aspects.

**Classification of Errors Using Major/Minor Error Guidelines**   Drawing inspiration from the MQM translation human rating system and InstructScore prompting template (Freitag et al., 2021; Xu et al., 2023c), we classified translation errors as either major or minor. This classification helped GPT-4 in assigning more consistent scores to each error, countering its instability with numerical judgments (Lu et al., 2023).

**Adopting a 0.5 to 5 Scale for Scoring Error Severity**   Initially, we used an integer scale from 1 to 5 for error penalty scores. While effective in translation tasks, this scale is less effective in tasks like summarization, data2text, and instruction-following. Our experiments demonstrated that a more nuanced scoring scale ranging from 0.5 to 5 yielded better correlation across all tasks

## A.2 Creation of evaluation aspects

With the assistance of GPT-4, we have carefully curated the evaluation aspects of each task that are mutually exclusive and collectively exhaustive. The steps include:

- Step 1: We prompt GPT-4 to output 20 candidate aspects for each task.

- Step 2: We ask GPT-4 to summarize these aspects into 3 to 5 general aspects for this task.

- Step 3: We ask GPT-4 to generate detailed definition and 5 specific error types under each aspect.

In each step, we check the reasonability of GPT-4's response and make necessary modifications to the responses, including summarized error aspects, error definition, and error types, to make them clear, concise, and typical. Example prompts used in each step are show in Table 21.

## A.3 TIGERScore prompt template

---

You are evaluating errors in a model-generated output for a given instruction.
Instruction:
${generation_instruction}
${input_context}

Model-generated Output:
${hypothesis_output}

For each error you give in the response, please also elaborate the following information:
- error location (the words that are wrong in the output)
- error aspect it belongs to.
- explanation why it's an error, and the correction suggestions.
- severity of the error ("Major" or "Minor").
- reduction of score (between 0.5 and 5 given the severity of the error)

Your evaluation output:

---

Table 9: The prompt template used for the fine-tuning and inference of TIGERSCORE

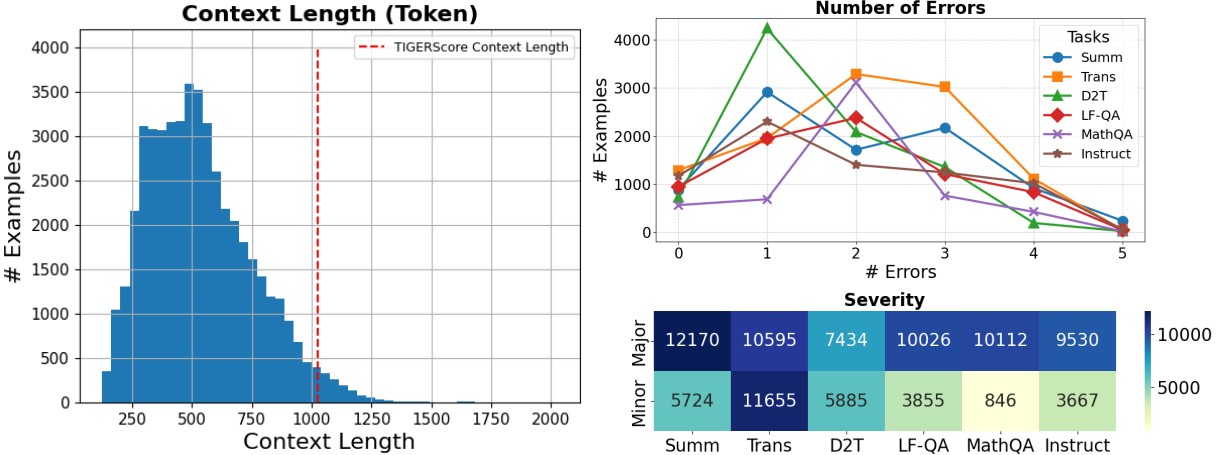

Figure 4: Distribution analysis of MetricInstruct training data for the context length, number of errors, and error severities. The left figure shows the context length distribution. The right-top plot illustrates the distribution of the per-instance number of errors. The right-bottom figure illustrates the counts of "Major" and "Minor" errors for all six tasks

## A.4    MetricInstruct data statistics analysis

In order to find the secret of the success of MetricInstruct, we conducted a deep analysis of its inherent statistics, as shown in Figure 4. we first count the distribution of data length through the Llama tokenizer. The results show that more than 90% of data has a length lower than 1024. Due to the lack of GPU resources for fine-tuning in the longer context length scenario, this distribution further demonstrates the reasonability of using 1024 as the context length of TIGERScore.

Furthermore, we examined the incidence of errors per data instance across various tasks. The figure illustrates that each task contains most instances with the number of errors being 1 or 2 and fewer instances are considered perfect and very erroneous, reflecting a naturally occurring distribution. We contend that such a balanced distribution is crucial for the model's performance—it helps in reducing fabricated errors in correct outputs and aids in the precise identification of minor and major errors in outputs that are partially correct or entirely incorrect.

Additionally, we categorized errors as 'Major' or 'Minor' and quantified their occurrences. Our analysis reveals that tasks of a more subjective nature, such as translation and summarization, tend to have a higher frequency of minor errors. Contrastingly, in tasks like MathQA, the predominance of major errors is in alignment with the expectation that mathematical inaccuracies are generally more critical."

## A.5    Gold scores for correlation computation

Human preference scores are necessary to conduct the correlation analysis. Those datasets with official system outputs released are usually accompanied by systematic human preference scores, like the WMT-MQM score for the translation task. However, these scores are not available for tasks like long-form QA, MathQA, and instruction-following, where we need to create on our own.

Therefore, we here introduce what human preference scores we have used to conduct the evaluation experiments. For summarization, data2text, and story generation tasks, we use their official human ratings from multiple aspects of their released outputs. For translation, we use the official WMT-22 MQM scores as the gold scores. For MathQA, we simply use the accuracy (1 or 0) as the gold preferences. For instruct-following, we use human ratings from the hugging face community of a dataset where 500 instances are sampled from LIMA and AlpacaEvla. For Long-form QA, we use the powerful GPT-4 to perform pairwise comparisons for them and count the winning times as the way to rank them, which is similar to how Jiang et al. (2023) constructs MixInstruct.

## A.6 Evaluation aspects for all tasks

| Task | Aspect | Definition |
|---|---|---|
| Summ | Relevance | The degree to which the summarized output accurately reflects the key points of the input text. |
| | Fact Consistency | If the facts in the summary are consistent with the facts in the original text. |
| | Coherence | Pertains to the logical and meaningful arrangement of ideas in the summary. |
| | Fluency | Reviews the model-generated output's use of language, including grammar, punctuation, and vocabulary that affect the quality of the sentences. |
| Trans | Accuracy | The degree to which the translated text adheres to the original text, maintaining the same meaning, context and cultural nuances. |
| | Fluency | How naturally the translation reads in the target language. |
| | Terminology | The appropriate use of specific terms and jargon related to a particular field or industry. |
| | Style Matching | Translator's ability to maintain the same style, tone, and voice as the original text. Example error types include: |
| D2T | Accuracy | Deals with the correctness of the information presented by the output. |
| | Logical Coherence | How well the output transforms structured data into a comprehensible, logical, and engaging text. |
| | Fluency | Reviews the model-generated output's use of language, including grammar, punctuation, and vocabulary that affect the quality of the sentences. |
| LF-QA | Accuracy | Evaluates the factual correctness of the answer. |
| | Completeness | Evaluates if the answer leaves out any critical parts or details that were asked in the question. |
| | Informativeness | Assesses the quality of the response in terms of how helpful it is for the user to understand the answer. |
| | Clarity | Assesses the readability and understandability of the response. |
| MathQA | Problem Understanding | Assesses how well the output accurately comprehend the text-based description of the math problem. |
| | Problem Formulation | Involves translating the problem from a textual form into a mathematical equation or set of equations that can be solved. |
| | Computing Accuracy | Assesses the output's ability to perform the mathematical operations accurately to arrive at the correct solution. |
| | Solution Interpretation | Involves the how well the output correctly interpret the solution of the problem in the context of the original problem statement. |
| Instruct | Comprehension | Evaluates how well the output understands the given instruction. |
| | Accuracy | Measures the correctness of the output in relation to the instruction and the paired input context. |
| | Informativeness | Assesses the relevancy and usefulness of the information provided by the output. |
| | Coherence | Evaluates how logically the output flows and connects. |

Table 10: Definitions of evaluation aspects of TIGERSCORE for the 6 text generation task.

## A.7 Prompting templates

Source: ${input_context}
Reference: ${reference_output}
Output:${hypothesis_output}
Based on the given Source and Reference, please evaluate the quality of summary(Output) written for the input text. Please score the summarization with 0.5 to 5 for aspects below. Then, identify the major and minor errors in this output for the $task task. There may be multiple errors or no error in the output. Here are the aspects you need to focus on: ${aspects_descriptions}

Table 11: Prompting templates for summarization task

Translation Instruction: ${generation_instruction}
Source Text: ${input_context}
${reference_output}
Model-generated Translation: ${hypothesis_output}
Please identify and categorize the errors in the model-generated translation as either Major or Minor. Major errors significantly impact the task, while Minor errors are subjective and represent minor imperfections.
When identifying errors, do not solely rely on the reference translation for comparison. Provide explanations as an expert in the task domain, without explicitly mentioning the reference output.

Table 12: Prompting templates for translation task

Task instruction:{generation_instruction}
Source: $ {input_context}
$ {reference_output}
Output: $ {hypothesis_output}
Based on the given source and reference, identify the major and minor errors in this Output for the data to text task, which is to $generation_instruction. Note that Major errors refer to actual errors that affects the task severely, may change the meaning of the output, and Minor errors refer to smaller imperfections, and purely subjective opinions about the output. There may be multiple errors or no error in the output.

Table 13: Prompting templates for data2text task

${generation_instruction}
${input_context}
The correct solution is:
${reference_output}
A model-generated solution is: ${hypothesis_output}
Please identify all the errors in this output considering the following aspects: ${aspects_list}

Table 14: Prompting templates for mathQA task

Source: ${input_context}
${reference_output}
Output: ${hypothesis_output}
Based on the given Source and reference, identify the major and minor errors in this Output for the ${task} task, which is to ${generation_instruction}. Note that Major errors refer to actual errors that affects the task severely, may change the meaning of the output, and Minor errors refer to smaller imperfections, and purely subjective opinions about the output. You should check about ${aspects_descriptions}.There may be multiple errors or no error in the output.

Table 15: Prompting templates for long-form QA task

${generation_instruction_and_source}
${reference_output}
Output: ${hypothesis_output}
Based on the given Source and reference, identify the major and minor errors in this Output for the ${task} task. Note that Major errors refer to actual errors that affects the task severely, may change the meaning of the output, and Minor errors refer to smaller imperfections, and purely subjective opinions about the output. You should check about ${aspects_descriptions}.There may be multiple errors or no error in the output.

Table 16: Prompting templates for instruction-following task

For each error identified in your response, please provide the following information in a specific JSON format:
- Error Location: The substring in the Output that contains the error.
- Error Aspect: Choose only one from ${aspects_list}.
- Explanation: Describe why the identified issue is an error, and offer suggestions for correction. Explain as an expert in the task domain, without explicitly mentioning the reference output.
- Severity: Classify the error as "Major" or "Minor".
- Score Reduction: Assign a reduction score between 0.5 and 5, considering the severity of the error.
JSON Format for Output:
- If there are no errors:
{"errors": {}}
- If there are errors:
{"errors": {
"error_1": {
"error_location": "...",
"error_aspect": "...",
"explanation": "...",
"severity": "...",
"score_reduction": ...
},
...
}}

Table 17: Prompt template to format the response into json format

You are generating an output given the following instruction and context:
${generation_instruction}
${input_context}

The correct output is: ${reference_output}

Please generate an incorrect output for the given instruction and context by modifying the
correct output. The incorrect output should contain the following errors:
${error_requirements}
For each error, give me the
- error location (the substring that is wrong in the generated incorrect output)
- error aspect
- explanation (the generic error type description, why it's an error, and the correction suggestions)
- severity ("major" or "minor")
- score reduction (an integer between 1 to 5 given the severity of the error)

Output format:
Generated incorrect output:

Error location 1:
Error aspect 1:
Explanation 1:
Severity 1:
Score reduction 1:
...

Table 18: Prompt template to generate data in the synthetic channel for all 6 tasks.

Instruction:
${instruction}
${input}

A ground-truth response:
${output}

A model will be asked to respond to this instruction. However, that response might contain errors in various aspects.

Please first output 5 possible error aspects if a model is asked to generate a response for the above instruction. The error aspects don't have to be one of the above aspects and can be any aspect that you think is reasonable for this instruction.

Then generate an incorrect response contains up to ${num_errors} errors of these aspects. Each error corresponds to one of the aspect.
The incorrect response should mimic style the real-generation of a model.

Then give an analysis of these errors. For each error, give me the
- error location (the substring that is wrong in the generated incorrect output)
- error aspect
- explanation (the generic error type description, why it's an error, and the correction suggestions)
- severity ("major" or "minor")
- score reduction (an integer between 1 to 5 given the severity of the error)

Output format:
Generated incorrect output:

Error location 1:
Error aspect 1:
Explanation 1:
Severity 1:
Score reduction 1:
...

Table 19: Prompt template to generate data with free-form aspects in the synthetic channel for instruction-following task. Data generated from this template aims to improve TIGERScore's generalization ability on the unseen error cases and enhance the robustness.

${instruction}
${input}

Model-generated output:
${output}

An error analysis provided:
${error_analysis}

Is the error analysis reasonable? Answer me "yes" or "no" only.

Table 20: Prompt template used to check whether the error analysis synthesized is reasonable or not.

You are evaluating a model-generated output for the mathematical word problem.
There might be multiple errors in the output and they can focus on different evaluation aspects.
Please list 20 evaluation aspects for mathematical word problems and their definitions.

Please summarize the above-provided aspects into 3 to 5
general aspects that are mutually exclusive and collectively exhaustive.

Mathematical word problems are a type of math task that requires model
to read and understand a text-based description of a math situation and
then formulate and solve a mathematical equation to solve the problem.
Now you are evaluating a model-generated output for a mathematical word problem.
you might want to design a few aspects where errors are identified in the outputs
could be attributed to these aspects.
These aspects could be
"Problem Understanding", "Problem Formulation",
"Computing Accuracy", "Solution Interpretation",
provide the definition of each aspect above and 5 specific error types for each aspect.
Note that these aspects are used to evaluate a single output instead of a system.

Table 21: An example prompt when creating the template of mathQA

Task instruction: ${instruction}
Source: ${input}
Reference output: ${reference_output}
Model-generated output:
${output}
An error analysis provided:
${error_analysis}
Does error analysis include factual errors or hallucinations of the model-generated output?

Table 22: The prompt template we used to identify whether there are hallucinated contents or factor errors in TIGERSCORE's response

### A.8 Human evaluation case study

**Task Instruction** - Answer the following ambiguous factoid question by introducing additional knowledge, clarifying the relationship between multiple possible answers (if any) and resolving the ambiguity

**Input Context** - When did the song stayin alive come out?

**A model-generated response** - The song "Stayin' Alive" is a disco hit produced by the Bee Gees and was included in the soundtrack of the film "Saturday Night Fever" in 1977. The song was released as a single in December 16, 1577 and quickly became a popular dance anthem, topping the charts in several countries. However, it is important to note that there is another song called "Streetlife" by The Crusaders that also became popular around the same time and has a similar tempo to "Stayin'Alive", which could be confusing.

Identified errors in the model-
generated response from TigerScore-
The model-generated output contains 2 errors, with a total score reduction of 6.
Error location 1: "The song "Stayin' Alive" is a disco hit produced by the Bee Gees and was included
in the soundtrack of the film "Saturday Night Fever" in 1977. The song was released as a single
in December 16, 1577"
Error aspect 1: Accuracy
Explanation 1: The error is a factual inaccuracy. The song "Stayin' Alive" was not released in 1577,
but in 1977. The correct information should be provided to ensure the accuracy of the answer.
Severity 1: Major
Score reduction 1: 4
Error location 2: "However, it is important to note that there is another song called "Streetlife" by
The Crusaders that also became popular around the same time and has a similar tempo to
"Stayin'Alive", which could be confusing."
Error aspect 2: Informativeness
Explanation 2: This is an example of irrelevance. While it is true that there is a song called "Streetlife"
by The Crusaders that became popular around the same time as "Stayin' Alive",
it is not relevant to the question asked. The user asked for the release date of "Stayin' Alive",
not for similar songs that may have been popular at the same time.
The answer should stick to the information requested.
Severity 2: Minor
Score reduction 2: 2

Table 23: A case study of our human evaluation results for long-form QA task, the main body. Human evaluation questions and example answers are shown in Table 24

1. Please first read through all the contents above. Which identified errors are not reasonable?

All the errors are reasonable and the identified errors are true errors

Error 1

Error 2

2. Is there any significant errors that not identified?

No. There are no missing identified errors

Yes. But the missed error is minor and does not affects the overall evaluation

Yes. And the missed error seems to a major error, but most other major errors are also identified.

Yes. And the evaluation completely misses all the major errors and does not make sense.

3. Please rate the quality of the TigerScore evaluation results for the model-generated output from 1 to 5

1. The identified errors by TigerScore does not make sense and help at all

2. TigerScore manages to identify some errors, though few explanations of them are reasonable, you can still find something useful in the outputs.

3. TigerScore is able to idenfitify part of major errors and some of explanations are helpful, despite some errors still being missed.

4. TigerScore is able to identify most of the errors some most of the explanations are helpful. The missing errors are minor and are dispensable.

5. TigerScore identified all the errors and all the explanations make sense and are very helpful.

4. Does the explantion provide helpful revision suggestions?

No. Not at all

Yes. But they are not helpful at all.

Yes. And some of them are helpful.

Yes. And most of them are helpful despite some small mistakes

Yes. And they are all helpful enough for human to revise this model-generated output.

Table 24: Human evaluations questions and corresponding answers from a human rater for the instance in Table 23

## A.9 Correlation results on Pearson and Spearman

| Tasks→ | Summ | Trans | D2T | LF-QA | MathQA | Instrct | Story-Gen | Average |
|---|---|---|---|---|---|---|---|---|
| GPT-based Metrics | | | | | | | | |
| GPT-3.5-turbo (zero-shot) | **45.53** | **43.77** | **47.76** | 29.84 | **61.26** | 15.36 | 7.80 | 35.90 |
| GPT-4 (zero-shot) | 40.75 | 33.92 | 46.83 | **49.30** | 54.98 | **60.45** | **37.74** | **46.28** |
| Reference-based Metrics | | | | | | | | |
| BLEU | 11.66 | 17.47 | 34.29 | 18.21 | 18.12 | 29.47 | -0.64 | 18.37 |
| ROUGE-2f | 16.03 | 16.26 | 35.85 | 19.66 | 20.69 | 33.49 | 2.88 | 20.69 |
| InstructScore | 27.40 | 51.55 | 47.28 | 20.59 | 0.36 | 20.98 | 12.81 | 25.85 |
| GPTScore-ref | 13.47 | 21.05 | 48.70 | 33.40 | 18.22 | 29.66 | 18.94 | 26.20 |
| BARTScore-cnn (hypo-ref) | 16.67 | 23.56 | 45.08 | 32.78 | **23.09** | 26.57 | 27.61 | 27.91 |
| BARTScore-para (hypo-ref) | 19.73 | 29.04 | 47.89 | 32.70 | 17.33 | 30.20 | 17.76 | 27.81 |
| BERTScore | 26.26 | 37.65 | 48.22 | 26.39 | 11.19 | 45.58 | 4.08 | 28.48 |
| BLEURT | 17.27 | 43.00 | **54.32** | 34.26 | 3.98 | 39.15 | 27.89 | 31.41 |
| UniEval (summ) | **53.22** | 23.11 | 51.14 | **36.95** | 17.69 | 30.87 | **44.88** | 36.84 |
| COMET-22 | 35.32 | **58.46** | 43.82 | 36.79 | -5.58 | **49.68** | 40.12 | **36.94** |
| Reference-free Metrics | | | | | | | | |
| BARTScore-para (src-hypo) | 43.11 | 6.96 | 37.82 | 29.86 | -0.41 | 19.37 | 19.99 | 22.38 |
| BARTScore-cnn (src-hypo) | 39.72 | 9.53 | 45.43 | 41.48 | 3.28 | 34.97 | 33.51 | 29.70 |
| Llama-2-13b-chat-0-shot | 29.59 | 9.09 | 41.32 | 21.67 | 2.80 | 22.71 | 21.13 | 21.19 |
| COMETKiwi | 14.22 | **50.91** | 23.63 | 22.59 | -13.35 | 34.46 | 19.12 | 21.65 |
| GPTScore-src | 41.71 | 6.82 | 41.19 | 39.79 | 13.99 | 27.59 | 23.22 | 27.76 |
| TigerScore-7B | 43.95 | 37.70 | 49.13 | **46.10** | 21.77 | 38.26 | 39.90 | 39.54 |
| TigerScore-13B | **44.21** | 41.54 | **52.87** | 44.76 | **24.41** | **47.52** | **47.66** | **43.28** |
| Δ (ours - best reference-free) | +1 | -9 | +7 | +5 | +10 | +20 | +14 | +13 |
| Δ (ours - best reference-based) | -9 | -17 | -2 | +9 | +1 | -2 | +3 | +6 |

Table 25: The **Pearson** correlation results of all the baseline metrics and TIGERSCORE on the evaluation datasets shown in Table 3. For each task, the metric with the highest correlation to average performance is highlighted in bold.

| Tasks→ | Summ | Trans | D2T | LF-QA | MathQA | Instrct | Story-Gen | Average |
|---|---|---|---|---|---|---|---|---|
| GPT-based Metrics | | | | | | | | |
| GPT-3.5-turbo (zero-shot) | **38.50** | 40.53 | 40.20 | 29.33 | **66.46** | 23.20 | 4.77 | 34.71 |
| GPT-4 (zero-shot) | 36.46 | **43.87** | **44.04** | **48.95** | 51.71 | **58.53** | **32.48** | **45.15** |
| Reference-based Metrics | | | | | | | | |
| BLEU | 11.98 | 19.73 | 33.29 | 11.38 | 21.12 | **46.61** | -1.17 | 20.42 |
| ROUGE-2f | 14.53 | 17.83 | 35.49 | 16.83 | 22.12 | 44.56 | 2.34 | 21.96 |
| InstructScore | 26.33 | 50.43 | 38.54 | 21.62 | -4.15 | 16.19 | 16.13 | 23.58 |
| GPTScore-ref | 14.73 | 24.95 | 39.42 | 31.60 | 18.20 | 33.14 | 18.24 | 25.75 |
| BARTScore-cnn(hypo-ref) | 13.64 | 28.53 | 36.12 | 29.57 | **23.35** | 32.49 | 26.64 | 27.19 |
| BARTScore-para (hypo-ref) | 17.18 | 33.72 | 40.79 | 28.94 | 17.27 | 34.47 | 17.43 | 27.11 |
| BERTScore | 23.67 | 42.41 | 43.75 | 25.60 | 11.53 | 45.77 | 2.88 | 27.95 |
| BLEURT | 17.30 | 48.41 | **48.76** | 33.26 | 3.53 | 36.46 | 27.52 | 30.75 |
| UniEval(summ) | **47.52** | 21.90 | 38.38 | **41.83** | 19.78 | 16.02 | **44.46** | 32.84 |
| COMET-22 | 33.75 | **56.35** | 33.92 | 35.28 | -5.53 | 46.13 | 39.20 | **34.16** |
| Reference-free Metrics | | | | | | | | |
| BARTScore-para (src-hypo) | **38.68** | 9.60 | 32.26 | 26.86 | -2.70 | 5.92 | 20.55 | 18.74 |
| BARTScore-cnn (src-hypo) | 35.50 | 12.83 | 34.33 | 40.96 | 1.50 | 25.43 | 33.48 | 26.29 |
| Llama-2-13b-chat-0-shot | 28.53 | 14.38 | 29.24 | 19.91 | 1.08 | 21.37 | 26.78 | 20.18 |
| COMETKiwi | 16.27 | **48.48** | 27.90 | 18.05 | -11.48 | 34.86 | 18.47 | 21.79 |
| GPTScore-src | 37.41 | 8.90 | 28.82 | 39.48 | 14.25 | 26.46 | 23.91 | 25.61 |
| TIGERScore-7B (ours) | 35.11 | 41.50 | 42.39 | **47.11** | 21.23 | 43.57 | 39.26 | 38.60 |
| TIGERScore-13B (ours) | 36.81 | 44.99 | **45.88** | 46.22 | **23.32** | **47.03** | **46.36** | **41.52** |
| Δ (ours - best reference-free) | -2 | -3 | +12 | +5 | +9 | +14 | +13 | +16 |
| Δ (ours - best reference-based) | -9 | -11 | -3 | +5 | -0 | +0 | +2 | +7 |

Table 26: The **Spearman** correlation results of all the baseline metrics and TIGERScore on the evaluation datasets shown in Table 3. For each task, the metric with the highest correlation to average performance is highlighted in bold.

