# OpenReview forum: "TIGERScore: Towards Building Explainable Metric for All Text Generation Tasks"
_TMLR — Accepted by TMLR_

### Review · Reviewer_hP4V · 2024-03-25

**Summary Of Contributions:**

The paper introduces TIGERScore, an innovative metric for evaluating text generation tasks that prioritizes explainability and reference-free assessment.

**Audience:**

Yes

**Claims And Evidence:**

Yes

**Requested Changes:**

Take comprehensive comparison on GPT-based metrics with the TIGERScore on both accuracy and efficiency.

**Strengths And Weaknesses:**

Strengths:
1. The human evaluation of TIGERScore's explanations provides valuable insights into the metric's interpretability and usefulness, validating its effectiveness in generating understandable error analyses.
2. The paper's claim regarding the potential for TIGERScore to serve as a universal metric for text generation evaluation.

Weaknesses:
While the paper claims near parity with the GPT-4 evaluator, a deeper analysis is warranted, especially in comparison to GPT-based few-shot or fine-tuned models. Table 4 indicates that GPT-based zero-shot evaluation outperforms TIGERScore on certain tasks. This observation prompts further investigation into how TIGERScore fares against GPT-based few-shot or fine-tuned models.

---

> ### Author Response · Authors · 2024-04-16
> **Response to Reviewer hP4V**
>
> Firstly, we are glad to see you recognize TIGERScore's value of interpretability and usefulness. Your proposed weakness is also significant for us to improve our paper. We are willing to solve your concern here with detailed explanations:
>
> 1. Firstly, thanks for your raised concerns about the results in Table 4. By using “comparable performance with GPT-4”, **we originally intended to state that TIGERScore has a close performance with GPT-4**. In terms of average performance, GPT-4 achieves **36.38**, while TIGERScore archives **33.73**, which is only about **3 points behind**. We also add justification for the “comparable performance” in the experiments analysis, “What’s exciting to note is that TIGERScore13b can achieve comparable correlation performance with GPT-4 (zero-shot), even higher on summarization, translation, data2text, and story generation” in second 4.3. That’s why we use the term “comparable performance” in the statement in terms of accuracy.
>
> 2. In terms of efficiency and cost, GPT-4 is a way larger model than our 13B TIGERScore. It’s also close-source. Every time prompting GPT-4 requires the billings concerning the number of tokens used. However, **TIGERScore is an open-source model and can be simply deployed locally**. We have also provided multiple deployment options for users to use TIGERScore, including **VLLM**, **4-bit quantization**, **GGUF model for CPU-only inference with llamacpp support**. For example, **TIGERScore deployed with VLLM takes only about 0.2 seconds to score each item, which is efficient in large-scale evaluation**.
>
> 3. **Compared to other GPT-based models (excluding GPT-4), TIGERScore apparently achieves way higher correlation performance as shown in Table 4**. So we conclude TIGERScore is a good replacement for GPT-4 as a general-purpose evaluator considering that it’s an open-source, smaller model with fast inference support.
>
> We hope our explanation helps solve your concern and any further questions are welcomed. Thanks again for your valuable feedback!

---

### Review · Reviewer_hH1D · 2024-03-28

**Summary Of Contributions:**

This paper presents a trained metric, TIGERScore, which can be used for evaluating the ability of text generation models. The key characteristics of the proposed metric lie within that TIGERScore is reference-free and explainable. It can not only provide evaluation scores but also pinpoint the mistakes with informative error analysis. The metric is fine-tuned on LLaMA-2, based on meticulously curated instruction-tuning dataset MetricInstruct which covers 6 text generation tasks and 23 text generation datasets. Analysis shows that  TIGERScore correlates well with human ratings on 5 held-in datasets, 2 held-out datasets. It is evident that TIGERScore can achieve the open-source SoTA correlation with human ratings across these datasets and almost approaches GPT-4 evaluator. This paper also provides a thorough discussion on the error cases and hallucination analysis. Overall, this work makes a solid contribution of an effective metric for evaluating text generation.

**Audience:**

Yes

**Broader Impact Concerns:**

The ethics statements are informative.

**Claims And Evidence:**

Yes

**Requested Changes:**

A few clarifications. Please see the weaknesses.

**Strengths And Weaknesses:**

Strengths:

1. This paper presents a valuable metric for evaluating model performance among a wide range of text generation tasks. It has the advantage of generality, explainability, and free from reference.

2. The paper is generally well written and easy to follow. The motivation is clear and the background is presented in detail. The dataset and approach are also described clearly.

3. The analysis is thorough. When reading the paper, I have raised a few question. While reading forward, it is good to see most of concerns have been well addressed.

Weaknesses:

My major concern is that some implementation details are not clarified.

1. In Section 2.2, how the penalty score is determined? It seems that the penalty score is directly given by prompting GPT-4. However, as there would be bias in GPT-4 output, it remains unclear whether the absolute value of the score is convincing or not.

2. This work adopts Kendall’s correlation with human ratings. It is better to justify the choice. Besides, it is interesting to see whether the results may vary with different correlation measures.

3. The key claim may be incorrect. This work positions the scope as the "evaluation of natural language generation tasks". However, the actual contribution seems to be something like the evaluation of text generation "models" instead of "tasks".

Minor Comments:

There are a few writing mistakes. A more careful proofreading is needed. Here are some examples.

1. The caption of Figure 1: "moel-generated" -> "model-generated"

2. Section 3.2: a running-on sentence,"representing 2 main components of MetricInstruct"

---

> ### Author Response · Authors · 2024-04-16
> **Response to Reviewer hH1D**
>
> We are grateful that you recognize TIGERScore as a valuable metric and glad that the paper is deemed as easy to follow and understand. Your raised weaknesses are also valuable feedbacks to our paper and we are willing to address these concerns one by one as follows:
>
> For weakness 1:
> 1. Firstly, the prompting process to get the final synthesized data involves 2 turns of conversation for each item. For the first turn, we apply different templates for different tasks in table 11-16. For the second turn, we apply the template shown in table 17 with the context history in the first turn. As shown in table 17, We request GPT-4 to first generate an additional severity level along with the score. We have 2 severity types, which are Major and Minor. For Major errors, the penalty score should be between [2.5,5]. For Minor errors, the penalty score should be between (0, 2.5). **We have manually removed all those items where GPT-4 generated absolution scores inconsistent with the severity level. In this way, we think it makes the absolute value more reliable by eliminating outlier scores**.
>
> 2. On the other hand, there are previous studies on whether GPT-4 is a good and consistent evaluator when generating scores: [https://arxiv.org/abs/2308.02575](https://arxiv.org/abs/2308.02575). This paper conducted intraclass correlation coefficients (ICC), showing that GPT-4’s evaluation scores are consistently more than 95% of the time if experiments are replicated. **These findings also add support for the reliability of using generated absolute values as the scores**.
>
> For weakness 2:
> 1. **Due to the page limitation, we already put our Pearson and Spearman correlation results in the appendix as supplementary experiment results to Kendall’s results**. As shown in the table, TIGERScore still achieves the best performance compared to other reference-free metrics in most of the tasks. It also achieved the overall best average performance, only behind GPT-4. **These metrics indicate the same conclusion with Kendall**, that TIGERScore is a powerful and useful general-purpose, reference-free, explainable metric for all text generation tasks
>
> For weakness 3:
> 1. Thank you for pointing out the potential confusion points. As shown in the title, by using “evaluation for all text generation tasks”, we mean that **you can easily evaluate the output for any task by simply changing the task instruction in the prompt template of TIGERScore**. Indeed, we do not aim to evaluate the tasks to see whether the task is good or not. The starting sentence in the introduction paragraph: “Evaluation of natural language generation tasks”, might be a little confusing. A more appropriate term should be “evaluating LLM's performance on any natural language generation tasks”. We have changed it into “Evaluation for natural language generation tasks” by replacing “of” with “for”, indicating that we care about better evaluation methods for different tasks, instead of the evaluation of the task itself.
>
>
> We hope these clarification help solve your concerns. Further questions are welcomed and we are willing to respond.
>
> We have also modified the raised typos and are grateful for the reviewer's time to find out these typos. They are all great improvements to the paper quality!

---

### Review · Reviewer_2PF2 · 2024-04-02

**Summary Of Contributions:**

This paper proposes TigerScore, which is a text generation evaluation metric, which follows human instructions and provides the evaluation scores with the error analysis.

**Audience:**

Yes

**Claims And Evidence:**

No

**Requested Changes:**

1. The authors could better elaborate the contributions and novelty in Introduction and related work.

2. Please make sure all the comparisons are fair.

**Strengths And Weaknesses:**

**Strengths:**

The proposed method is evaluated on 6 tasks across 23 datasets, providing a solid experimental validation.


**Weaknesses:**

1. The instruction following mechanism and providing (instruction, input, system output → error analysis) input-output framework as well as the format of error analysis report are highly similar to InstructScore [1], leading to the novelty of this paper limited.

[1] Instructscore: Towards explainable text generation evaluation with automatic feedback. EMNLP 2023.

2. The performance of InstructScore is pretty bad, while in their paper, their performance is good. Did the author train their datasets on all baselines which are trained metrics?  If not, I don't think this is a fair comparison.

---

> ### Author Response · Authors · 2024-04-16
> **Response to Reviewer 2PF2**
>
> Thanks for recognizing our experiments as comprehensive and solid. And we are also thankful for your raised weaknesses and concerns. We here respond to your concerns mentioned as the weakness as follows and hope it can help you better understand TIGERScore's contributions.
>
> For weakness 1:
> 1. **As stated in the introduction, TIGERScore aims to provide a solution to a reference-free, explainable automatic metric for all text generation tasks. On the contrary, InstructScore is reference-based and domain-specific**. It’s stated in the InstructScore paper section 5.1 that, *“We train a separate checkpoint for each evaluation scenario, resulting in four checkpoints in total”*, please check out https://huggingface.co/xu1998hz and you will multiple checkpoints and the main one is for machine translation. That means you need to train a brand-new checkpoint for each new task with the new data for the domain adaptation.
>
> 2. **However, TIGERScore is trained only once**. Then it’s ready for the evaluation of any task by simply changing the instruction in the prompt. It does not require training a new checkpoint specifically with additional data, thus making it a general-purpose and cross-domain evaluator. We believe deeply that **reference-free**, **cross-domain evaluation**, and **instruction-guided** are essential characteristics of a good evaluator and a key novelty compared to the InstructScore paper. **These characteristics are never shown in the InstructScore, making it a less powerful model than TIGERScore**.
>
> 3. Again, an important distinction is that our model can do explainable instruction following evaluation, which is the most popular and practical task for LLMs. **None of the existing metrics are designed for it**. Our paper’s main focus is mainly on understanding the LLM evaluator’s generalization to new tasks and new instructions. In contrast, InstructScore is only for task-specific evaluation without considering generalization.
>
> For weakness 2:
> 1. As stated above, InstructScore has *“a separate checkpoint for each task”*.  **However, in our evaluation setting, each model is evaluated using the same checkpoint, thus making it an unfair comparison if we train a separate checkpoint of InstructScore for each of our evaluation tasks**. Therefore, we use InstructScore’s main checkpoint fine-tuned for the MT evaluation scenario for all 7 tasks. Our reported InstructScore Kendall correlation on WMT-22 (zh-en) is **40.44, consistent with their number of 40.3, reported in the InstructScore paper**. Therefore, we believe the comparison with InstructScore is fair and consistent.
>
> 2. *"Did the author train their datasets on all baselines which are trained metrics?”*. The curated dataset, MetricInstruct, is one of our major contributions to our paper. We don’t think it’s fair and doable to train each baseline model on MetricInstruct for comparison. **In a similar case, the InstructScore work also did not train new metrics in their reported baselines on their datasets**, even though BERTScore, BARTScore, COMET, BLEURT, and SEScore2 are all trainable metrics. Besides, InstructScore has a separate checkpoint for each evaluation scenario, which is task-specific. **Training a new model using the InstructScore paradigm on our curated datasets makes it a different model from the original InstructScore**.
>
> 3. Given these points, we contend in the end that our current comparison is fair and consistent.
>
> We hope these statements help solve your concern, and assist you in better understanding that there is the key difference between TIGERScore and InstructScore work. Any further questions are welcomed and we are willing to clarify.

---

### Review · Reviewer_upZt · 2024-04-03

**Summary Of Contributions:**

This work investigates the automatic evaluation of natural language generation (NLG) tasks. It introduces a trainable reference-free evaluator/metric *TIGERScore* that provides not only the final evaluation score but also the error analysis of the generated texts when assessing the performance of a NLG model. Specifically, *TIGERScore* is a LLaMA-2 model tuned on a manually crafted dataset, namely *MetricInstruct*, where the samples are selected from 23 existing datasets across 6 NLG tasks, with additional annotated error analysis generated by prompting GPT-4. Experimental results show the effectiveness of the proposed *TIGERScore* on 5 held-in datasets and 2 held-out datasets. A human evaluation is also conducted to validate the accuracy of error analysis generated by *TIGERScore*.

**Audience:**

Yes

**Claims And Evidence:**

No

**Requested Changes:**

Please address the concerns in weaknesses.

**Strengths And Weaknesses:**

=== Strengths ===
- Overall the paper reads good and is easy to follow.
- The idea of introducing a reference-free metric that also provides explanations (i.e., error analysis) for NLG tasks is wise.
- The constructed dataset *MetricInstruct* could be a good contribution to the research community if open sourced.

=== Weaknesses ===

The main concerns are two folds:
- The experimental setup is unclear and not justified, which could be biased and lead to unfair comparison.
- Unconvincing human evaluation, failing to reflect the actual performance of *TIGERScore* in generating error analysis

Below are the itemized questions:
1. In section 2.2, do we need to assume the availability of natural language instruction $I$ for each
 task? How is it determined in this work?
2. As presented in Table 3, only a small number of input texts $x$ are sampled from each dataset. How is this selection determined? One potential concern is that human selection could be biased and lead to unfair comparison. Are the comparison results in Table 4 sensitive to the sample selection strategy and/or number of testing samples? Do we have any concerns about increasing the test data size to make it more representative?
3. In section 4.4, how many human experts are employed to evaluate the generated error analysis? And what is their agreement rate? The statistics in Table 5 suggests the entire human evaluation is based only on one human evaluator, is that correct? If so, then the human evaluation may not be convincing enough.
4. In section 4.5, using GPT-4 to determine if the generated error analysis of *TIGERScore* contained hallucinations or factual inaccuracies sounds problematic given TIGERScore is trained on GPT-4 generated data -- it's known that GPT-4 as judge can have biased evaluation for texts generated by itself. Therefore, the hallucinations analysis of TIGERScore outputs may not be accurate. Do we have any idea to mitigate this bias?

=== Typos ===
- Caption of Figure 1: "... **moel-generated** output ..." $\rightarrow$ "... **model-generated** output ..."
- First line of Section 4.3: "**6 held-in** tasks and **1 held-out** task" $\rightarrow$ "**5 held-in** datasets and **2 held-out** tasks"

---

> ### Author Response · Authors · 2024-04-16
> **Response to Reviewer upZt**
>
> Thanks for recognizing our paper's strengths and raising valuable concerns for our paper. We have here responded and clarified your concerns in the mentioned questions:
>
> 1.  Yes, to evaluate an item, an instruction is needed for every term.
>    - During the training, for each task, there are multiple datasets with different specific instructions. Since these instructions might not have been previously available, we have our authors write the instructions for each dataset manually. We have designed a dataset-instruction mapping where each dataset has a human-written instruction for that dataset. **They are either written according to the original paper descriptions or through the investigation of the task dataset description.**.
>    - For another part of the training dataset (10k), which comes from the alpaca-52k, we instead use their original instruction as the task instruction. These instructions are more diverse thus enhancing TIGERScore to be an instruction-following evaluator.
>    - During the evaluation, the instructions are also written before.
>    - During the actual inference, the instruction can be custom instruction that describes a task.
> 2. We did not sample a small number of input texts from each dataset. Table 3 presents evaluation datasets, and the statistics reported are the original dataset statistics. Since each item in the evaluation dataset has multiple outputs generated from different systems, the number of samples actually means the total number of outputs.
> 3. During the human evaluation, we indeed only have 1 human expert to annotate due to the limitation of resources and funding. However, as stated in the ethical statements, these annotators are recruited from the famous Prolific platform, where the human annotator’s background and ability are verified through the system.
> 4. We respond to this weakness in 2 points:
>     - We admit the possible bias might be brought by using GPT-4 as a judge. However, since GPT-4 is the SoTA LLM with excellent ability, we believe the results are still trustworthy. Besides, the evaluation using GPT-4 is actually much easier than using GPT-4 to synthesize data. During the evaluation, GPT-4 is only required to judge whether the errors contain fact errors or hallucinations given both the source text, reference output, incorrect output, and the TIGERScore generated errors. We think easier tasks usually are less affected by the model’s bias, thus making the output more trustworthy.
>     - Besides, also in section 4.5, we perform an experiment called “alignment on no-error outputs”, where no other LLMs are required to do the evaluations. This experiment can also prove that TIGERScore does not hallucinate a lot since about 77.21% of the no-error outputs get less than 2 score deductions.
>
> We hope this information helps you better understand TIGERScore's contributions as well as solve your concerns.
>
> We have also corrected the typos in the latest paper revisions. Thanks a lot for pointing out these typos!

---

### Decision · Action_Editor_ZsC3 · 2024-05-07

**Recommendation:** Accept as is

**Comment:**

The paper introduces a trained, reference-free, and explainable metric, TIGERScore, for a wide range of text generation tasks. The reviewers generally feel that the paper is well-written and contains solid and thorough experiments. After author's rebuttal, the concerns raised by the reviewers have been mostly resolved. The remaining concerns are: (1) the human evaluation is relatively small-scale; (2) the similarity to InstructScore; (3) some parts of the writing are unclear (e.g., experiment setup and implementation details). The AE believes that these concerns are quite minor and don't offset the merits of the paper. However, the authors are encouraged to consider reviewers' comments when preparing the final version.

**Audience:**

Yes

**Claims And Evidence:**

Yes